# Open-Vocabulary Video Scene Graph Generation via Union-aware Semantic Alignment

## ABSTRACT

Video Scene Graph Generation (VidSGG) plays a crucial role in various visual-language tasks by providing accessible structured visual relation knowledge. However, the requirement of annotating all categories of prevailing VidSGG methods limits their application in real-world scenarios. Despite the popular VLMs facilitating preliminary exploration of open-vocabulary VidSGG tasks, the correspondence between visual union regions and relation predicates is usually ignored. Therefore, we propose an Open-vocabulary VidSGG framework named Union-Aware Semantic Alignment Network (UASAN) to explore the alignment between visual union regions and relation predicate concepts in the same semantic space. Specifically, a visual refiner is designed to acquire open-vocabulary knowledge and the ability to bridge different modalities. To achieve better alignment, we first design a semantic-aware context encoder to achieve a comprehensive semantic interaction between object trajectories, visual union regions, and trajectory motion information to obtain semantic-aware union region representations. Then, a union-relation alignment decoder is utilized to generate the discriminative relation token for each union region for final relation prediction. Extensive experimental results on two benchmark datasets show that our UASAN achieves significant performance over existing methods, which also verifies the necessity of modeling union region-predicate alignment in the VidSGG pipeline. Code is available in **Supplementary Material**.

## CCS CONCEPTS

• **Computing methodologies → Scene understanding**.

## KEYWORDS

Open-vocabulary Learning, Video Scene Graph Generation, Scene Understanding

## 1 INTRODUCTION

Video Scene Graph Generation (VidSGG) task aims to detect and localize the visual relationships between different entity trajectories in a given video, constructing the relationships as relation triplets in the form of *<subject-predicate-object>*. It serves a crucial

Permission to make digital or hard copies of all or part of this work for personal or classroom use is granted without fee provided that copies are not made or distributed for profit or commercial advantage and that copies bear this notice and the full citation on the first page. Copyrights for components of this work owned by others than the author(s) must be honored. Abstracting with credit is permitted. To copy otherwise, or republish, to post on servers or to redistribute to lists, requires prior specific permission and/or a fee. Request permissions from permissions@acm.org.
*ACM MM, 2024, Melbourne, Australia*
© 2024 Copyright held by the owner/author(s). Publication rights licensed to ACM.
ACM ISBN 978-x-xxxx-xxxx-x/YY/MM
https://doi.org/10.1145/nnnnnnn.nnnnnnn

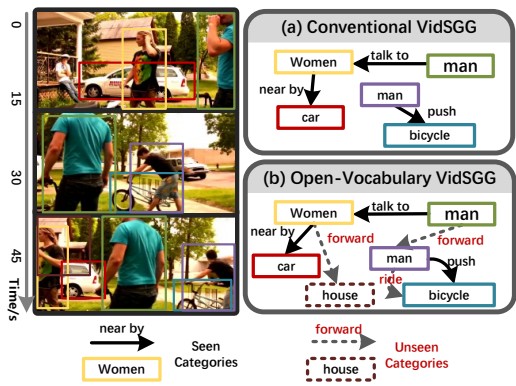

**Figure 1: Open-vocabulary setting. Conventional closed-set VidSGG frameworks only recognize the objects and relation predicates been seen during model training, while open-vocabulary VidSGG approaches can be generalized to unseen object and predicate categories.**

role in various visual comprehensive tasks, such as visual question answering [3, 15, 16], video retrieval [6, 7], and video captioning [25, 48], by furnishing structured knowledge to enhance video understanding.

Despite achieving impressive performance, existing VidSGG frameworks [8, 24, 26, 32–34, 49] remain constrained to recognizing objects and predicting visual relations within closed-set scenarios, which entails that the categories of objects and relation predicates are pre-defined and manually annotated. However, such the closed-set model training process prevent current VidSGG frameworks from being employed in real-world scenarios, due to their inclusion of various visual object or relation concepts that do not appear or are unseen in the model training set. When encountering these novel categories, current VidSGG methods are likely to fail to recognize or classify them into known categories as shown in Figure 1. Meanwhile, the movement of objects in real world scenarios over time makes the relationship between visual objects blurred and complicated, which also makes annotating more difficult. Moreover, due to the expensive and time-consuming labor costs for annotating, collect all categories from a real-world scenario is also not accessible. Therefore, it is crucial to address how to imbue a VidSGG model with the generalization ability to recognize novel categories when only being trained on limited categories.

To reduce the need for annotating novel categories and improve the generalization of the models, open-vocabulary learning has been explored in the object detection field, named open-vocabulary object detection (OVD) [10, 11, 39, 44]. Specifically, the aim of open-vocabulary learning is to train a model with annotations on a part of classes (*i.e.*, base classes), and generalize it to unseen classes (*i.e.*, novel classes) during inference. Inspired by this, there have

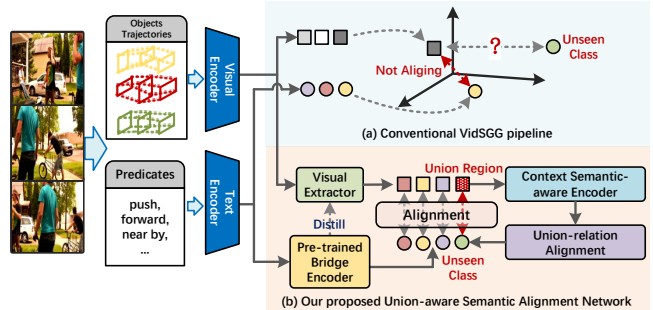

**Figure 2: Overview of the conventional VidSGG pipeline vs. our proposed OV-VidSGG method. Conventional VidSGG methods are trained within a closed-set and don't have the ability to recognize unseen object and predicate categories. They typically rely on the aligned visual and textual object features from pre-trained VLMs for relation prediction, while the alignment between subject-object pairs and predicate concepts is insufficient and unreliable. Contrarily, our proposed UASAN framework explores the alignment between the visual union regions and relation predicate concepts for better relation prediction performance.**

also been some preliminary studies on the open-vocabulary scene graph generation task (Ov-SGG) [9, 12, 47] in recent years.

Benefiting from the encyclopedic knowledge of popular vision-language models (VLMs) [20, 22, 28], such as CLIP [28], existing Ov-SGG methods [9, 12, 47] can easily recognize novel objects and explore novel relationships through the object-level alignment between visual objects and textual category labels. However, these Ov-SGG methods excessively rely on the aligned visual and textual object features provided by pre-trained VLMs for model learning, while ignoring the alignment between visual relation regions and relation predicate concepts. The visual relation region is represented as the union region of two objects, and is usually regarded as a type of assistant complement for relation prediction. In fact, most existing methods simply fuse such union region representations with object features, such as concatenating them with object features [47], for relation classification, which is shown in Figure 2(a). It is obvious that only using subject-object pair features (*e.g.*, *woman* and *house* in Figure 1) to be aligned with textual predicate embeddings (*e.g.*, *near by* in Figure 1) is insufficient for relation prediction. Though some recent closed-set scene graph generation works [50] have perceived the correspondence between visual union region information and visual relation concepts, they do not yet explicitly model the correspondence between them and relation concepts and still directly conduct alignment in the semantic space, which results in a lack of comprehensive interaction and correspondence between visual semantic information. As indicated in previous research [13, 43], VLMs (*e.g.*, CLIP [28]) still encounter challenges in performing compositional scene understanding, such as inter-object relation, which results in incomplete and unreliable alignment between subject-object pair representations and relation predicate representations and leads to an ambiguous relation prediction.

Therefore, we propose a novel Ov-VidSGG framework named **Union-Aware Semantic Alignment Network** (UASAN) to explicitly model the alignment between visual union regions and relation predicates in the same semantic space for joint feature fusion, and the framework is shown Figure 2(b). Specifically, we design a visual refiner guided by a bridge encoder to transfer the open-vocabulary knowledge and the ability to bridge the modality gap into our framework. It facilitates our model paying more attention to visual-relevant semantic information, which results in a sequence consisting of visual-aware subject, object, union region, and motion information representations. After that, we design a semantic-aware context encoder to achieve comprehensive interaction within the constructed sequence for obtaining the semantic-aware representations. Then we propose a union-relation alignment decoder to generate union-aware tokens based on the encoded sequence for final relation predicate prediction.

In summary, our contributions in this paper are as follows:

(1) We propose an open-vocabulary video scene graph generation method named Union-Aware Semantic Alignment Network (UASAN), which explicitly explores the alignment between the object trajectory union regions and the relation predicates for a more comprehensive relation prediction.

(2) Our proposed UASAN consists of three cooperative components: (1) A visual refiner is designed to transfer the knowledge and ability into our framework for obtaining visual-aware representations; (2) a semantic-aware context encoder is utilized to achieve comprehensive semantic-aware interaction based on the representations from our visual refiner; and (3) a union-relation alignment decoder is proposed to integrate semantic-aware representations for generating discriminative relation tokens for final prediction.

(3) Extensive experiments on two benchmark datasets, *i.e.*, Vid-VRD and VidOR datasets, demonstrate the effectiveness of our proposed framework.

## 2 RELATED WORK

**Open Vocabulary setting in SGG task.** Scene Graph Generation (SGG) task [8, 17, 33, 34, 40, 41] aims to generate visual relation triplets in a given image or video, and provide the structure visual relation information for benefiting various downstream multimodal tasks. While existing SGG methods have achieved impressive performance on prominent public datasets, they are limited to predicting visual objects and their relations within a closed-set environment. The aforementioned constraint significantly hinders the practicality of SGG methods in real-world scenarios, due to they rely on training with known classes of objects and relations.

Recently, the closed-set SGG has been extended to unseen classes through efforts made in the zero-shot setting [23, 42], where the triplets for inference are not seen in the training set. Moreover, He *et al.* [12] proposes a more challenging open-vocabulary setting for image-based SGG task (Ov-SGG). To be specific, in Ov-SGG, the model is trained only utilizing the objects from a pre-defined set of seen object categories (or base categories), subsequently predicting relationships among unseen object categories (or novel categories). Both seen and unseen sets are subsets of the open-vocabulary object class set. It means that not only the object combinations but also the object categories themselves may not

be seen during model training. Therefore, [12] utilize among of region-caption pairs for pre-training a visual-relation model, and finetune the relation model with prompt learning strategy. Besides, [12] also define a more challenging setting named general Ov-SGG, in which the predicate categories set are divided into novel set and base set. The former contains novel predicates during model inference that are not seen in training. Given the advantages conferred by pre-trained visual-language models (*i.e.*, CLIP and GLIP), there is an inclination towards tapping into the ability of these VLMs for better visual or semantic relevant works. Inspired by the trend, Zhang *et al.* [47] explore the pre-trained visual semantic space (VSS) and propose a novel SGG model named $VS^3$ to transfer the language-image knowledge for benefiting Ov-SGG. Besides, due to VLMs (*e.g.*, CLIP) struggle to distinguish between different relation types, Li *et al.* [21] integrate LLMs (*e.g.*, GPT [1]) into their model to generate detailed composition descriptions based on a chain-of-thought strategy for a better relation prediction.

Different form the aforementioned SGG frameworks that are based on image, Gao *et al.* [9] is the first to explore the Open-vocabulary Video Scene Graph Generation setting, and propose a novel framework named Relation Prompt Learning framework (Re-Pro), where compositional prompt is utilized to capturing complex spatial-temporal information for predicate representation learning. Although RePro have make preliminary attempts, the potential benefit from the correspondence between visual union regions and relations is still not considered. Moreover, some recent scene graph generation works [50] have token the visual union region information into consideration, they do not yet explicitly model the alignment between them and relation concepts, and are not appropriate to open-vocabulary setting. Different from them, we explore such alignment in open-vocabulary setting for achieving a robust open-vocabulary VidSGG framework.

**Video Scene Graph Generation.** Video Scene Graph Generation (VidSGG) task[2, 4, 8, 24, 26, 27, 33–37], aiming to detect and localize the visual relationships between different entity trajectories in a given video, has been widely used in various visual comprehensive task. Shang *et al.* [34] is the first to propose the VidSGG task with releasing a dataset named ImageNet-VidVRD, and propose a novel VidSGG framework named VidVRD. Inspired by Vid-VRD, Qian *et al.* [27] and Tsai *et al.* [37] focus on exploring the spatio-temporal information with a graph structure for relation prediction. Moreover, [36] design a Target Adaptive Context Aggregation Network to capture context information for each subject-object pair. Different from them, Su *et al.* [35] propose a novel Multiple Hypothesis Association framework, which pays more attention to maintains the constructed multiple relations for selecting accurate ones. Moreover, Gao *et al.* [8] decompose the VidSGG pipeline, and propose a classification-then-grounding framework assisted by a video temporal grounding module for triplet localization. Although existing VidSGG approaches have achieved great performance, they are still limited to a closed-set training process, which prevents them from being applied in real-world scenarios. Therefore, in this paper, we propose an open-vocabulary VidSGG framework to improve the generalization of our model for recognizing unseen categories.

## 3 THE PROPOSED APPROACH

Figure 3 illustrates our proposed framework, aiming to achieve open-vocabulary video scene graph generation by modeling the alignment between visual union regions and relation predicate concepts. We first design a heuristic structure, which we called bridge encoder, to access the open-vocabulary knowledge and the ability bridging modality gap for achieving object trajectory classification. Then we explore the alignment between union region and relation predicate concept. Specifically, we design a visual refiner to transfer such knowledge and ability from the bridge encoder into our framework by distillation. Relying on the ability bridging different modalities distilled from bridge encoder, the object trajectory features extracted by visual refiner also contains textual semantic characteristic. Therefore, we concatenate the representations of objects, union region, and motion information into a sequence to simulate textual relation triples, and a context semantic-aware encoder is designed to achieve comprehensive semantic understanding among these triplets. Then we propose a union-region alignment decoder to generate union-aware relation tokens. Finally, we incorporate the relation tokens with the union and motion information for relation predicate prediction.

### 3.1 Preliminary

**Problem Definition.** Given a video $V$, video scene graph generation (VidSGG) aims to detect the visual entities and their relationships in the form of relation triplet $< s, p, o >$, where $s$ and $o$ are the class labels of subject and object, and $p$ is the class label of predicate. To conduct open-vocabulary VidSGG setting, following [9] we divide the categories of the objects and the predicates collected from all the annotations into base split and novel split. Specifically, we denote the base and novel object categories as: $O_{base}$ with the number of $N_{base}^O$ and $O_{nvoel}$ with the number of $N_{novel}^O$, respectively, as well as $\mathcal{P}_{base}$ and $\mathcal{P}_{novel}$ denoted as the base and novel predicate categories. We train our method with the triplet samples only containing base object and predicate categories, while evaluating it on both novel and all categories.

**Object Trajectory Generation and Feature Extraction.** We utilize the same features as it in [9]. Specifically, a pre-trained object trajectory detector is employed to generate sets of class-agnostic trajectories from a given video. The detected object trajectories are denoted as $\mathcal{T} = \{T_i\}_{i=1}^N$. For each trajectory $T_i$, it consists of a bounding box sequence $T_i = \{\mathbf{b}_j\}_{j=1}^M$, where $\mathbf{b}_j$ is the bounding box, and $\mathbf{b}_j \in \mathbb{R}^4$. For each trajectory $T_i$, we use a pre-trained ViT [5] for feature extraction, and denote it as $\mathbf{f}_i \in \mathbb{R}^d$. Note that a video is typically cut into short video segments at first for computational simplicity during model training. To explore the alignment between visual union regions and predicate concepts, we extract union regions for the subject-object pairs in each video segment, where we construct $N_P$ pairs totally, where $N_P = N * (N - 1)$. The union region is the union of the bounding boxes of subject and the ones of object for each pair, and we also use pre-trained ViT to extract union region features, which is denoted as $\mathbf{U} \in \mathbb{R}^{N_P \times d}$. Moreover, we also collect motion information for each subject-object trajectory pair $< T_s, T_o >$ as the same in [30, 33] to make our model notice the relative position and the motion trend between objects in relation prediction,

Figure 3: Illustration of our proposed Union-Aware Semantic Alignment Network. We initially use a pre-trained ViT to extract visual representations of detected object trajectories and union regions, and obtain the word embeddings of the relation predicate concepts. For exploring the alignment between union regions and relation predicate concepts, a distillation strategy is firstly adopted to transfer the knowledge and ability from the pre-trained bridge encoder to our designed visual refiner. Then, we concatenate the subject, object, visual union regions, and motion information as a sequence to simulate a textual relation triplet, and construct a context semantic-aware encoder to achieve comprehensive semantic interaction. After that, a union-relation alignment decoder is designed to generate the union-aware relation tokens for further relation prediction. Finally, we aggregate the union-aware relation tokens, multiple union region features, and motion information together for a comprehensive relation prediction.

which is denoted as $\mathbf{f}_{s,o}^{mot}$. We utilize a mapping function, denoted as $\phi_{mot}$, to obtain the features of motion information for all subject-object pairs as: $\mathbf{F}_{mot} = \phi_{mot}(\{\mathbf{f}_{s,o}^{mot}, s, o \in [1, N], s \neq o\})$, and $\mathbf{F}_{mot} \in \mathbb{R}^{N_P \times L_{mot} \times d}$.

**Fine-tuning Object Trajectory Classification within the Open-vocabulary Setting.** Thanks to the popular VLMs, achieving impressive object classification is now more accessible. Inspired by the pre-trained BLIP-2, we achieve open-vocabulary trajectory classification by leveraging a Q-Former heuristic structure, referred to as bridge encoder, to access encyclopedic knowledge and alleviate the modality gap.

Specifically, following [9] we first allocate the category labels to the detected trajectories according to the Intersection over Union (IoU) with the ground-truth trajectories, considering only base object categories. Then, we bridge the visual and textual modalities as follows:

$$\mathbf{f}'_i = \text{BridgeEnc}(\mathbf{f}_i), \qquad (1)$$

$$\mathbf{O}_{base} = \text{BridgeEnc}(\text{WordEmb}(O_{base})), \qquad (2)$$

where WordEmb($\cdot$) means a word embedding project. Moreover, the probability of classifying trajectory $T_i$ to class $c \in O_{base}$ is as:

$$p_i^{traj}(c) = \frac{\exp(\cos(\mathbf{f}'_i, \mathbf{o}_c)/\tau)}{\sum_{c' \in O_{base}} \exp(\cos(\mathbf{f}'_i, \mathbf{o}_{c'})/\tau)} \qquad (3)$$

where $\mathbf{o}_c$ is the text embedding of the object category $c$.

According to [9], we define the trajectories with allocated labels as positive samples, while others are negative samples, and calculate the classification loss $\mathcal{L}_{traj}$.

## 3.2 Union-aware Semantic Alignment Learning

In consideration of the previous Ov-SGG methods excessively relying on the object-level alignment provided by pre-trained VLMs, we then explore the semantic correspondence between the visual union region and the relation predicate concept in this section.

**Knowledge Distillation Learning.** At first, we design a visual refiner to transfer the open-vocabulary knowledge from our pre-trained bridge encoder into our framework through a distillation strategy, which can facilitate our model paying more attention to visual-relevant semantic information. Specifically, we design a sequence of learnable visual concept tokens to obtain visual-aware concept features, and the tokens are denoted as $\mathbf{Q} = \{\mathbf{q}_1, \mathbf{q}_2, ..., \mathbf{q}_L\}$, where $L$ is the length of the sequence. To avoid the interference of noise contained in the detected object trajectories and reduce the costs during distillation, we utilize a small amount of manually annotated object trajectories, and also use pre-trained ViT to extract features, denoted as $\mathbf{F}_M = \{\mathbf{f}_m\}_{m=1}^M$.

To obtain visual-aware representations for each object trajectory, $\mathbf{Q}$ is extended to $\mathbf{Q}' \in \mathbb{R}^{M \times L \times d}$, and a attention module [38], denoted as $Attn(\cdot, \cdot, \cdot)$, is utilized for modeling the dependencies of the visual concept tokens, denoted as $\mathbf{V}_M = \text{Attn}(\mathbf{Q}', \mathbf{Q}', \mathbf{Q}')$. Then, a cross-attention-based module is used for comprehensive interaction between object trajectories and visual concept tokens, denoted as $\mathbf{F}_v = \text{Attn}(\mathbf{V}_M, \mathbf{F}_M, \mathbf{F}_M)$. After that, we use knowledge distillation strategy to transfer the knowledge into our framework. The distillation loss is as follows:

$$\mathcal{L}_{dis} = \|\mathbf{F}_v - \mathbf{F}'_M\|, \qquad (4)$$

where $\mathbf{F}'_M = \text{BridgeEnc}(\mathbf{F}_M)$.

**Semantic-Aware Context Encoder.** As similar as generating $\mathbf{F}_v$, we also obtain the visual-aware trajectory features $\hat{\mathbf{F}}$ and union region features $\hat{\mathbf{U}}$, where $\hat{\mathbf{F}} \in \mathbb{R}^{N \times L \times d}$ and $\hat{\mathbf{U}} \in \mathbb{R}^{N_P \times L \times d}$. Relying on the ability bridging different modalities of our designed visual refiner, the object trajectory features and union region features also contain textual semantic characteristic. Therefore, we utilize a semantic-aware context encoder (SACEncoder) to conduct comprehensive semantic interaction within the triplet to obtain semantic-aware representations. Specially, according to the constructed $N_P$ subject-object pairs before, we can obtain the features of the subject and object trajectories, denoted as $\mathbf{F}_S$ and $\mathbf{F}_O$, where $\mathbf{F}_S, \mathbf{F}_O \in \mathbb{R}^{N_P \times L \times d}$. Then, we concatenate the subject trajectories, object trajectories, the union region representation, and two types of motion features for simulating relation triplets, which also consist of subject, object, and predicate. Then, to reduce the influence of the order of the components in our sentence, we utilize a attention-based module to conduct interaction among each of them with a learnable token denoted as $\mathbf{f}_r$, and obtain the encoded "triplet" representations as follows:

$$\mathbf{F}_{sent} = [\mathbf{F}_S; \mathbf{F}_O; \hat{\mathbf{U}}; \mathbf{F}_{mot}], \tag{5}$$

$$\mathbf{F}_{enc} = \text{Norm}(\phi_{enc}(\text{Attn}(\mathbf{F}_{sent} + \mathbf{f}_r, \mathbf{F}_{sent} + \mathbf{f}_r, \mathbf{F}_{sent}))), \tag{6}$$

where $\phi_{enc}$ means a MLP, and Norm means normalization project. $\mathbf{F}_{mot} \in \mathbb{R}^{N_P \times L_{mot} \times d}$ is the extracted object motion features for all subject-object pairs.

**Union-relation Alignment Decoder.** After aggregating the context semantic from objects and motion information, we then need to generate the union-aware relation tokens for final relation prediction. Therefore, we design a union-relation alignment decoder (URADecoder), which is a multi-head attention module [38]:

$$\mathbf{t}_r = \text{MHAttn}(\mathbf{f}_r, \mathbf{F}_{enc}, \mathbf{F}_{enc}), \tag{7}$$

$$\mathbf{F}_{dec} = \text{Norm}(\mathbf{t}_r + \phi_{dec}(\mathbf{t}_r)), \tag{8}$$

where MHAttn is a multi-head attention module, and $\phi_{dec}$ means a MLP. With the assistance of the decoder, we can generate the union-aware relation tokens $\mathbf{F}_{dec}$ for further relation prediction.

**Prompt-based Union-relation Embeddings and Relation Predictor.** Different from traditional classification tasks, the number of categories to be predicted is uncertain during training and inference. Open-vocabulary model predicting relying on the pre-extracted predicate category embeddings. Therefore, we introduce the prompt learning strategy into our framework for accessing predicate category embeddings. Specifically, we utilize a sequence of learnable word embedding tokens as the prompt tokens for each predicate category, and extract predicate category embeddings by our pre-trained bridge encoder as follows:

$$\mathbf{W}_{rel}(\mathbf{c}) = [\mathbf{w}_1, \mathbf{w}_2, ..., \mathbf{w}_{L_{rel}}, \mathbf{c}], \tag{9}$$

$$\mathbf{r}_c^{rel} = \text{BridgeEnc}(\mathbf{W}_{rel}(\mathbf{c})), \tag{10}$$

where $\mathbf{w}_l(l \in 1, 2, ..., L_{rel})$ is a learnable word embedding. $\mathbf{c}$ is the predicate word embedding for class $c \in \mathcal{P}_{base}$ and $\mathbf{r}_c^{rel}$ is the category embedding.

After obtaining the union-aware tokens $\mathbf{F}_{dec}$, we can indeed use them for relation prediction. Additionally, to comprehensively consider the initial and semantic-aware union region information, we aggregate the multiple union region features, motion information, and union-aware tokens into a relation predictor for for a more robust predicate classification:

$$\mathbf{R} = \text{Norm}(\phi_{pre}(\mathbf{F}_{dec} + \mathbf{F}_{enc}^{[U]} + \hat{\mathbf{U}} + \mathbf{F}_{mot})), \tag{11}$$

where $\mathbf{F}_{enc}^{[U]}$ is the encoded union region representation obtained from $\mathbf{F}_{enc}$ and $\phi_{pre}$ means a MLP. For the each $< T_s, T_o >$ pair, the relation feature of them is denoted as $\mathbf{r}_{s,o} \in \mathbf{R}$. We compute the probability of relation predicate $c$ similar to Eq.(3), where we replace $\mathbf{f}'_i$ with $\mathbf{r}_{s,o}$ and replace the object category embeddings with the relation predicate embeddings $\mathbf{r}_c^{rel}$, where $c \in P_{base}$. The probability is denoted as $p_{s,o}^{rel}(c)$, and the relation prediction loss is similar to the classification loss in [9], denoted as $\mathcal{L}_{union}$.

### 3.3 Learning and Inference

**Model Training.** To complement union-level relation prediction process, we also take fine-grained subject-object pairs into relation prediction by a conventional strategy [9]. Specifically, we concatenate the subject and object trajectory features as the relation features, and leverage prompt learning strategy for generating predicate embeddings as similar to Eq.(9), denoted as $\mathbf{c}_c^{so}, c \in \mathcal{P}_{base}$, for final relation prediction. The loss is denoted as $\mathcal{L}_{fg}$.

Therefore, the final loss $\mathcal{L}$ for optimizing our framework is as follows:

$$\mathcal{L} = \mathcal{L}_{traj} + \mathcal{L}_{dis} + \mathcal{L}_{union} + \mathcal{L}_{fg}. \tag{12}$$

**Model Inference.** During model inference, we only utilize the detected class-agnostic object trajectories for trajectory classification and relation prediction. Note that during relation prediction, all object categories are utilized, while either novel or all predicate categories are used for prediction.

## 4 EXPERIMENTS

In this section, we conduct extensive experiments on public datasets and demonstrate the effectiveness of our proposed UASAN framework. Please refer to **Supplementary Material** for the code, trained model, detailed parameter settings and more experimental results.

### 4.1 Experimental Settings

**Datasets.** We evaluate our proposed method on the VidVRD [34] and VidOR [32] benchmarks: (1) The VidVRD dataset, comprising 1000 videos sourced from ILSVRC2016 VID [31], marks the pioneering effort in the realm of VidSGG task. It covers 35 object categories and 132 predicate categories. We follow the standard official splits: 800 videos for training and 200 videos for testing. The visual relations in each video are labeled by the relation triplets depicted as *<subject-predicate-object>*. (2) VidOR, a larger scale user-generated video dataset, consists of 10000 videos with a total length of 98.6 hours. VidOR dataset is dense annotated on 80 categories of objects and 50 categories of predicates with the same annotation format as VidVRD dataset. The whole dataset is divided into three splits: 7000 videos for training, 835 videos for validation, and 2165 videos for testing. Due to the fact that the validation set of VidOR is not publicly available, we evaluate our approach on the validation set.

**Evaluation Settings and Metrics.** Following [9], we evaluate our model on open-vocabulary object trajectory classification task

with Recall@K (R@K, K=5,10) metric, as the same evaluation proto-cols as in open-vocabulary object classification. For relation detec-tion evaluation, we evaluate our method on two traditional VidSGG tasks, i.e., Relation Detection (RelDet) and Relation Tagging (RelTag), with open-vocabulary setting. Specifically, the categories in dataset annotations are split into **base** categories and **novel** categories, where the former consists of the common categories while the latter consists of the rare ones. To comprehensively evaluate the performance of our model, two settings are adopted [9] for infer-ence: (1) Novel-split: the triplet samples with **all** object categories and **novel** predicate categories are utilized for model evaluation, and (2) All-split: the triplet samples with **all** object and predicate categories are used during inference. For RelDet task, Mean av-erage precision (mAP) and Recall@K (R@K, K=50,100) are used as evaluation metrics, while Precision@K (P@K, K=1,5,10) is used for RelTag task.

Moreover, three standard SGG evaluation tasks [45] are employed on VidSGG setting for further performance comparison: (1) Scene Graph Detection (SGDet), (2) Scene Graph Classification (SGCls), and (3) predicate classification (PredCls). We also use the mAP and R@K as the evaluation metrics for aforementioned tasks.

**Implementation Details.** The detected object trajectory data utilized in our work is the same as [9], where a Fast-RCNN [29]-based VinVL model [46] is used to detect object with bounding box for each video frame, and Seq-NMS is employed for class-agnostic object trajectory generation. We use a pre-trained ViT model [5] for trajectory feature extraction, and our bridge encoder is estab-lished based on a pre-trained Q-Former backbone [20]. To better adopt our model to the specific situation (i.e., VidVRD and VidOR), we use LoRA [14] for fine-tuning our bridge encoder to achieve open-vocabulary object trajectory classification. We also follow [27, 33, 34], generating visual relation triplets in short video seg-ments, and merge the same relations with greedy relation associa-tion algorithm proposed by [34]. For VidVRD dataset, the base split have 25 object categories and 71 predicate categories, while the novel split have 10 object categories and 61 predicate categories. For VidOR dataset, the base split consists of 50 object categories and 30 predicate categories, while the novel split contains 30 object categories and 20 predicate categories. The detailed splits please re-fer to [9]. The hidden size $d$ in our model is set to 512 and the length $L$ of the extracted features is set to 32. $L_{mot}$ is set to 2 and $L_{rel}$ is set to 10. We use the Adam optimizer [18] to train our model. The learning rate is set to $10^{-4}$ for VidVRD and $5\times10^{-5}$ for VidOR. The batch size is set to 8 for VidVRD and 4 for VidOR, and our model is trained 50 epochs on both VidVRD dataset and VidOR dataset. Considering the costs of model inference, we don't predict the vi-sual relations in the subject-object trajectory pairs where as least one of them predicated as background label. Due to space limita-tions, we place the experimental results on VidOR dataset in the **Supplementary Material**.

## 4.2 Evaluation on Open-Vocabulary Object Trajectory Classification

We compare our model with three baseline models on object tra-jectory classification task: ALPro [19], RePro [9], and BLIP-2 [20].

**Table 1: Performance comparison of open-vocabulary object trajectory classification on VidVRD dataset.**

| Models | VidVRD-novel | | VidVRD-base | | VidVRD-all | |
|--------|------|------|------|------|------|------|
| | R@5 | R@10 | R@5 | R@10 | R@5 | R@10 |
| ALPro | 41.38 | 53.81 | 40.21 | 61.97 | 38.07 | 55.14 |
| RePro | 46.34 | 50.42 | **79.34** | 81.81 | 63.31 | 65.62 |
| BLIP-2 | 59.90 | **72.97** | 46.84 | 58.38 | 50.41 | 62.51 |
| Ours | **68.70** | 70.79 | 78.68 | **82.32** | **73.51** | **76.39** |

Note that we only use the pre-traind ViT and Q-Former in BLIP-2 for object trajectory feature extraction and classification. And we report the results of our proposed model and other relevant methods for open-vocabulary object trajectory classification on the VidVRD dataset, which is shown in Table 1. We directly in-put the detected object trajectories and the textual classes into the encoders of two pre-trained VLMs, i.e., ALPro and BLIP-2, and cal-culate the similarity for classification. Different from them, RePro chooses to distill knowledge from ALPro with a MLP module, and achieves better classification performance than ALPro on all splits. We only finetune our bridge encoder, and achieve the SOTA per-formance. Specifically, our model outperforms BLIP by gains of 8.80% in terms of R@5 on novel split on VidVRD dataset, while it also achieves (23.10%, 13.88%) improvements on all split. When compared with RePro, though RepPro surpass our model by gains of 0.66% on R@5 on base split, we still outperform it on novel and all splits with averages of 15.93%. Note that RePro and our method are both trained only with base categories, our finetuned bridge encoder achieves better performance on novel split while main-tains the performance on base split. It demonstrate that the fine-tune strategy promotes our model better adopt to specific situation, thereby improving the classification performance of our model for both novel and base object categories.

## 4.3 Evaluation on Open-Vocabulary Scene Graph Generation

For generating a video sense graph in open-vocabulary setting, we separate the training processes of object trajectory classification and visual relation prediction. Therefore, the trajectory classifica-tion results are fixed during relation prediction.

**Comparison with SOTA Methods on Conventional VidSGG Setting.** We compared our proposed method with following meth-ods: MHA [35], VRD-SGTC [26], IVRD [24], BIG-C [8], and Re-Pro [9]. And the comparison results are shown in Table 2. From the results in Table 2, we have the observe that our proposed model achieves better performance on most of metrics, though only base-split data is used for model training. When compared with those trained conventionally methods (i.e., all object and predicate cate-gories are seen both in training and inference stages), we outper-forms BIG-C by gains of (5.90%, 6.27%, 7.94%) in terms of mAP, R@50 and R@100 on RelDet task, and we also achieve an improve-ment with an average of 6.37% on RelTag task. Moreover, we also

**Table 2: Comparison with state-of-the-arts on VidVRD datasets.**

| Methods | Training Data | Relation Detection | | | Relation Tagging | | |
|---------|---------------|--------|--------|--------|--------|--------|--------|
| | | mAP | R@50 | R@100 | P@1 | P@5 | P@10 |
| VRD-SGTC | base+novel | 18.38% | 11.21% | 13.69% | 60.00% | 43.10% | 32.24% |
| MHA | base+novel | 19.03% | 9.53% | 10.38% | 57.50% | 41.40% | 29.45% |
| IVRD | base+novel | 22.97% | 12.40% | 14.46% | **68.83%** | **49.87%** | 35.57% |
| BIG-C | base+novel | 17.67% | 9.63% | 11.29% | 56.00% | 43.80% | 32.85% |
| RePro | base | 21.33% | 12.92% | 15.94% | 59.00% | 41.09% | 28.87% |
| RePro* | base | 19.66% | 12.60% | 16.11% | 60.50% | 43.90% | 32.08% |
| UASAN | base | **23.57%** | **15.90%** | **19.23%** | 65.50% | 49.50% | **36.77%** |

surpass IVRD with an average of 2.96% on SGDet task. When compared with MHA and VRD-SGTC, our proposed method significantly improves over them by (6.37%, 4.69%) and (8.85%,5.54%) under R@50 and R@100 on RelDet, respectively. Such large improvements demonstrate that our proposed union-relation alignment framework have the ability to recognize predicate categories better though trained with only apart of categories. When compared with the conventional VidSGG approaches, we also achieve significant improvements.

Our proposed UASAN consistently achieves the best performance on all metrics compared to RePro [9]. Specifically, we outperform RePro by gains of (2.24%,2.98%,3.29%) on mAP, R@50 and R@100 metrics on RelDet task. In addition, when evaluated on RelTag task, UASAN also surpasses RePro with an average of 7.60%. Moreover, to facilitate a fair comparison the relation prediction performance of our model with RePro, we design a RePro variant (denoted as RePro* in Table 2) in which the object classifier is replaced by the classifier pre-trained in our framework, aiming to mitigate the influence of the performance of open-vocabulary object trajectory classification. We can observe that the variant achieves comparable performance with RePro. Specifically, RePro outperforms RePro* by gains of (1.67%, 0.32%) on mAP and R@50 metrics on RelDet task, while the variant achieves improvements of (1.50%,2.81%,3.27%) on P@1, P@5 and P@10 metrics. The comparison results investigate that the relation prediction performance of our framework is more relevant to our designed alignment strategy between union regions and predicate concepts, and our proposed modules (*i.e.*, visual refiner, semantic-aware context encoder and union-relation alignment encoder) also facilitate our model towards a comprehensive and robust relation prediction.

**Comparison with SOTAs on Open-vocabulary VidSGG Setting.** Comparison in the open-vocabulary setting, we train our model with base-split, and evaluate it with novel and all-split. The results on VidVRD dataset are summarized in Table 3. From Table 3 we can observe that our proposed method already achieves the best results on almost all metrics. Specifically, when evaluated on novel-split, we outperform VidVRD-II by gains of (7.48%, 4.79%, 5.96%) on SGDet task. We also surpass it with an average of 10.32% on mAP, R@50 and R@100 metrics on SGCls task. When compared with RePro, our proposed method surpasses it with (4.95%, 0.50%, 1.83%) in terms of mAP, R@50 and R@100 on SGDet task on novel-split. Moreover, when evaluated on SGCls and PredCls tasks, our proposed framework outperforms RePro with improvements of (4.18%,

3.97%, 5.30%) and (4.88%, 3.81%, 2.65%) for novel-split, respectively. For all-split, UASAN also achieves clear margin gains on SGDet (*e.g.*, 23.57% vs. 21.33% on mAP and 15.90% vs. 12.92% on R@50). The superior performance of our model demonstrates the necessity of exploring the alignment between visual union regions and relation predicate concepts, which facilitates the generalization ability on unseen categories while maintaining the recognition ability for seen categories during model training.

## 4.4 Ablation Studies

To investigate the effectiveness of each component of our model, we conduct ablation studies on VidVRD dataset. We implement three variants of our model as follows: (1) A conventional pipeline is implemented, where we concatenate subject and object features as the relation tokens and directly conduct predicate classification. This variant is denoted as Model $\mathcal{A}$ make it our baseline. (2) We introduce the union regions and the object motion information of the trajectories into our framework to generate relation tokens for predicate classification, denoted as Model $\mathcal{B}$. (3) We add the designed context semantic-aware encoder to Model $\mathcal{B}$ to achieve comprehensive context semantic understanding, and the encoded representations are utilized for relation prediction. This variant is denoted as Model $C$. And the model using all designed modules is denoted as full-model. Table 4 presents the performance of each model variant.

**Exploring Facilitation of Pre-trained Models.** To explore the influence brought by the employed pre-trained models (ALPro utilized in RePro and Bridge Encoder utilized in our framework), we implement the Model $\mathcal{A}$, which only considers the alignment between predicate concept embeddings and the subject-object pair features, which are the concatenated subject and object features as in RePro. In other words, we replace the ALPro in RePro with pre-traind Bridge Encoder. According to the results in Table 4 we can find that Model $\mathcal{A}$ achieves comparable performance with RePro. Specifically, when evaluated on novel-split, RePro outperforms Model $\mathcal{A}$ by gains of (2.31%, 1.31%) on mAP, R@50 and R@100 on SGDet task, while Model $\mathcal{A}$ achieves improvements of (0.63%, 1.00%) on R@50 and R@100 metric under all-split.

**Effectiveness of Exploring Union Region Modeling.** We develop the Model $\mathcal{B}$ to verify the necessity of modeling the alignment between union regions and relation predicate concepts. In Model $\mathcal{B}$, we simply design a sequence of learnable word tokens, and generate union-level predicate embeddings. Then, the union

**Table 3: Comparison of existing Open-vocabulary VidSGG methods on VidVRD dataset.**

| Split | Methods | SGDet | | | SGCls | | | PredCls | | |
|-------|---------|-------|-----|------|-------|-----|------|---------|-----|------|
| | | mAP | R@50 | R@100 | mAP | R@50 | R@100 | mAP | R@50 | R@100 |
| Novel | ALPro | 1.05% | 3.14% | 4.62% | 3.69% | 7.27% | 8.92% | 4.09% | 9.42% | 10.41% |
| | VidVRD-II | 3.57% | 8.59% | 12.39% | 5.70% | 13.22% | 18.34% | 7.35% | 18.84% | 26.44% |
| | RePro | 6.10% | 13.38% | 16.52% | 10.32% | 19.17% | 25.28% | 12.74% | 25.12% | 33.88% |
| | UASAN | **11.05%** | **13.88%** | **18.35%** | **14.50%** | **23.14%** | **30.58%** | **17.62%** | **28.93%** | **36.53%** |
| All | ALPro | 3.20% | 2.62% | 3.18% | 3.92% | 3.88% | 4.75% | 4.97% | 4.50% | 5.79% |
| | VidVRD-II | 12.74% | 9.90% | 12.59% | 17.26% | 14.93% | 19.68% | 19.73% | 18.17% | 24.90& |
| | RePro | 21.33% | 12.92% | 15.94% | 30.15% | 19.75% | 25.00% | 34.90% | 25.50% | 32.49% |
| | UASAN | **23.57%** | **15.90%** | **19.23%** | **32.24%** | **25.03%** | **31.07%** | **38.43%** | **30.01%** | **37.13%** |

**Table 4: Ablation studies of different components of UASAN on VidVRD dataset.**

| Split | Methods | Pair | Union | SACEnc | URADec | SGDet | | | SGCls | | | PredCls | | |
|-------|---------|------|-------|--------|--------|-------|------|-------|-------|------|-------|---------|------|-------|
| | | | | | | mAP | R@50 | R@100 | mAP | R@50 | R@100 | mAP | R@50 | R@100 |
| Novel | $\mathcal{A}$ | √ | × | × | × | 6.76% | 11.07% | 15.21% | 10.42% | 18.02% | 26.12% | 13.16% | 21.82% | 30.41% |
| | $\mathcal{B}$ | √ | √ | × | × | 7.69% | 12.56% | 16.53% | 11.12% | 22.98% | 27.93% | 13.40% | 27.60% | 34.05% |
| | $\mathcal{C}$ | √ | √ | √ | × | 10.09% | 12.56% | 16.03% | 12.81% | 21.32% | 26.61% | 14.44% | 26.28% | 32.89% |
| | full-model | √ | √ | √ | √ | **11.05%** | **13.88%** | **18.35%** | **14.50%** | **23.14%** | **30.58%** | **17.62%** | **28.93%** | **36.53%** |
| All | $\mathcal{A}$ | √ | × | × | × | 19.73% | 13.55% | 16.94% | 29.41% | 21.55% | 27.53% | 35.23% | 25.85% | 33.44% |
| | $\mathcal{B}$ | √ | √ | × | × | 21.13% | 15.35% | 18.43% | 31.06% | 24.20% | 29.97% | 37.07% | 28.93% | 35.59% |
| | $\mathcal{C}$ | √ | √ | √ | × | 22.49% | **16.19%** | 19.15% | 31.30% | 24.67% | 30.22% | 35.98% | 27.03% | 34.60% |
| | full-model | √ | √ | √ | √ | **23.57%** | 15.90% | **19.23%** | **32.24%** | **25.03%** | **31.07%** | **38.43%** | **30.01%** | **37.13%** |

region and the positional information are directly integrated into a predictor for relation prediction. It is obvious that modeling the correspondence of union regions and relations achieves improvements on almost all metrics. On novel-split, Model $\mathcal{B}$ outperforms Model $\mathcal{A}$ by gains of (0.93%, 1.49%, 1.32%) on mAP, R@50 and R@100 on SGDet task. Moreover, it also surpasses Mode $\mathcal{B}$ by an average of (2.48%, 3.22%) on SGCls and PredCls tasks on all-split. We can draw the conclusion that introducing the alignment between union regions and predicate concepts has a significant facilitating effect on improving model performance.

**Effectiveness of Perceiving Context Semantic.** Then we explore the effectiveness of our proposed semantic-aware context encoder. In Mode $\mathcal{B}$, we directly use the encoded semantic-aware representations for predicate classification. Comparing the results in Model $\mathcal{B}$ and Model $\mathcal{C}$ with novel-split, we can observe that though Model $\mathcal{B}$ outperforms Model $\mathcal{C}$ by gains of (1.66%, 1.32%, 1.32%, 1.16%) on R@50 and R@100 on SGCls and PredCls tasks, Model $\mathcal{C}$ achieves higher performance on the mAP metric on three tasks. Moreover, Model $\mathcal{C}$ surpasses Model $\mathcal{B}$ on most metrics when evaluated with all-split. Specifically, Model $\mathcal{C}$ outperforms Model $\mathcal{B}$ with improvements on mAP (22.49% vs. 21.13%,), R@50 (16.19% vs. 15.35%) and R@100 (19.15% vs. 18.43%) metrics on SGDet task. We speculate that it is because the designed semantic-aware context encoder promotes our model to pay more attention to the context semantic within subject-object pairs, and achieve better relation prediction performance.

**Effectiveness of Union-relation Alignment Decoder.** We finally investigate the benefits of our proposed union-relation alignment decoder. The results of our full-model are significantly better than Model $\mathcal{C}$. Specifically, full-model outperforms Model $\mathcal{C}$

by gains of (0.96%, 1.32%, 2.32%) on SGDet task under novel-split, which demonstrates the designed decoder brings our model stronger generalization to novel categories compared with only utilizing semantic-aware representations. In addition, full-model also achieves great improvements by gains of 1.08% on mAP metric when evaluated with the all-split. Additionally, clear improvements are also achieved on all metrics of SGCls and PredCls tasks. The results indicate the effectiveness of our designed decoder structure, which has the capability to integrate multiple representations and generate discriminative union-aware relation tokens for final robust relation prediction.

## 5 CONCLUSION

In this paper, we propose an open-vocabulary video scene graph generation framework named Union-Aware Semantic Alignment Network (UASAN), which explores the alignment between visual union regions and relation predicate concepts for more comprehensive and robust relation prediction. Specifically, we design a visual refiner to generate visual-aware representations for detected object trajectories and their union regions. Then we design a semantic-aware context encoder to obtain semantic-aware representations. After that we utilize a union-relation alignment decoder to generate discriminative union-aware relation tokens for final relation prediction. Extensive experimental results on VidVRD and VidOR benchmarks demonstrate our proposed UASAN outperforms almost all SOTA methods on Ov-VidSGG task. In the future, we aim to explore a lighter framework while maintaining comparable performance and application in various downstream tasks, such as VQA and video caption.

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
