# OpenReview forum: "Open-Vocabulary Video Scene Graph Generation via Union-aware Semantic Alignment"
_acmmm.org/ACMMM/2024/Conference — MM2024 Poster_

### Official Review · Reviewer_oACu · 2024-05-21

**Rating:** 5
**Confidence:** 4

**Summary:**

This paper addresses the alignment problem of visual regions and relationship predicates in the feature space for video scene graph generation. To achieve this alignment, the paper designs a semantic-aware context encoder and a union-relation alignment decoder. Extensive comparative experiments were conducted on the VidVRD and VidOR datasets.

**Strengths:**

1. This paper focuses on the alignment problem of visual regions and relationship predicates, which is crucial for scene graph generation and visual understanding. Compared to current object-based alignment methods, this paper provides a richer semantic understanding of visual information.
2. The figures in the paper are very clear and easy to understand, especially Figure 3, where the arrow indicators for the selection of Predicate category base are interesting.
3. There is a significant performance improvement on the VidVRD dataset.

**Limitations:**

1. The paper focuses on video scene graph generation, but the model structure does not include specific designs for understanding temporal content. Could this network be equally applicable to image scene graph tasks?
2. I suggest that the authors consider including the experiments on the vidor dataset in the main text, as the text mentions several times that experiments were conducted on the VidOR dataset, but all experimental tables in the later sections are based on VidVRD, which might confuse readers. I think many of the detailed explanations in the implementation details could instead be placed in the supplementary materials or the open-source code.
3. According to the experimental results, this method achieves significant performance improvements on the VidVRD dataset, but the improvements on the VidOR dataset are very limited. I am concerned whether this is due to the characteristics of the dataset's bias or an overfitting issue?
4. The motivation of this paper emphasizes the alignment of visual regions and relationship predicates. I suggest that the authors add visualization experiments on the alignment design. Comparing with object-only alignment methods could more intuitively demonstrate the importance of aligning visual regions and relation predicates.
5. I would like to know if the authors have explored the impact of different splits for base and novel categories.

**Suitability:**

3

---

### Official Review · Reviewer_kHK4 · 2024-05-23

**Rating:** 3
**Confidence:** 3

**Summary:**

The paper proposes an open-vocabulary video scene graph generation (VidSGG) framework named Union-Aware Semantic Alignment Network (UASAN). It aims to align visual union regions with relation predicates for better relation prediction in videos. The framework includes a visual refiner, a semantic-aware context encoder, and a union-relation alignment decoder. Experimental results on two benchmark datasets demonstrate the effectiveness of UASAN in comparison to existing methods.

**Strengths:**

1. The paper addresses the important challenge of extending video scene graph generation to an open-vocabulary setting, enabling recognition of unseen object and predicate categories. This improves the practical applicability compared to closed-set approaches.
2. The proposed UASAN framework explicitly models the alignment between visual union regions and relation predicate concepts, which the authors argue is important for robust relation prediction but under-explored in prior work.
3. Extensive experiments on VidVRD and VidOR datasets show significant performance improvements over state-of-the-art methods.

**Limitations:**

1. The novelty is limited since the proposed method relies heavily on pre-trained models like ViT, BLIP-2. The use of Knowledge Distillation seems like the engineering improvement, which lacks deep insights. While leveraging pre-training is common practice, it would be good to analyze the framework's robustness to different backbone choices. How much does performance degrade with smaller, more efficient backbones?  How much does performance degrade with multimodal large language models as backbones?
2. Related to the previous point, computational complexity and inference speed should be analyzed in the Experiment section. Runtime benchmarks and potential optimizations would strengthen the paper.
3. The experimental comparison focuses on open-vocabulary methods, but it would be good to include closed-set state-of-the-art results (even if not directly comparable) to contextualize the absolute performance.
4. Error analysis could yield additional insights - on which types of objects, relations, or videos does the approach still struggle? Qualitative visualizations of success and failure modes would help build intuition.
5. The paper does not sufficiently discuss how the model performs in varied real-world scenarios outside the benchmark datasets.
6. Please ensure consistent use of terminology throughout the paper to avoid confusion. For example, the different words of "visual refiner" and "visual feature extractor" in Figure 2 and Figure 3 could be confused.

**Suitability:**

3

---

### Official Review · Reviewer_LmPJ · 2024-06-04

**Rating:** 5
**Confidence:** 4

**Summary:**

This paper introduces an open-vocabulary VidSGG method that improves the alignment between visual union features and relation concepts. The authors finetune a Q-former (from BLIP-2) to bridge the visual-textural modalities for subject-object pairs and relation predicate categories. They also design a distillation strategy to transfer the pre-trained knowledge from Q-former to a visual refiner, and introduce semantic-aware context encoder with union-relation alignment decoder to enhance the feature alignment. Extensive experiments on VidVRD and VidOR datasets show the state-of-the-art performance and verify the effectiveness of each component.

**Strengths:**

-	The novelty and technical contributions are considerable.
 - This paper is the first one to introduce the semantic alignment between visual relation regions and predicate concepts for open-vocabulary VidSGG
 - This paper finds a novel and effective approach to improve the model’s open-vocabulary detection ability, i.e., finetuning the Q-former form BLIP-2 to bridge the visual-textural modalities for subject-object pairs and predicate categories.
-	The proposed UASAN model shows state-of-the-art open-vocabulary VidSGG performance on both VidVRD and VidOR benchmarks.
-	The experiments and ablation studies are comprehensive and all the modules of UASAN are verified to be effective.

**Limitations:**

-	Lack of insights for the union-aware semantic alignment mechanism in the open-vocabulary setting.
 - How does the union-aware semantic alignment prevent the model overfitting to the base classes?
 - Why the context semantic-aware and union-relation alignment are especially needed in the open-vocabulary settings? Does this also work well in conventional VidSGG setting?
-	The technical part (Sec.3) is a bit complicated. Some technical details are unclear. For example,
 - What is the detailed structure of visual refiner? Does it share the same structure as the Q-former and trained from scratch?
 - The queries $Q$ (Line 446) and $Q’$ (Line 453) are not illustrated in Figure 3. It’s a bit confusing whether they are queries for the Bridge Encoder or the Visual Refiner.
-	Some experiment settings are confusing.
 - In Sec.4.4, Line 497, the authors try to explore the influence of pretrained models (i.e., the ALPro in RePro vs. the Bridge Encoder in this paper). They say “we replace the ALPro in RePro with pre-traind Bridge Encoder”. However, the performance scores in Table 4 (e.g., 13.38% R@50 of SGDet) are exactly the same as those in the RePro’s paper (which is implemented with ALpro). Then, it’s unclear whether they re-implement RePro with their Bridge Encoder or implement their method with the ALPro in RePro.

-	Suggestions for the presentation of Figure 2.
 - For conventional VidSGG in sub-figure (a). In the closed setting, they typically train a classifier without using text encode. So, they do not suffer from the semantic alignment issue. The current sub-figure (a) might be misleading. A more appropriate comparison in (a) would be existing open-vocabulary VidSGG methods rather than conventional closed-set VidSGG methods

**Suitability:**

3

---

### Official Review · Reviewer_g8WY · 2024-06-06

**Rating:** 4
**Confidence:** 3

**Summary:**

The paper introduces the Union-Aware Semantic Alignment Network (UASAN), a  framework to tackle the challenges in Video Scene Graph Generation (VidSGG). UASAN aims to bridge this gap by exploring the alignment between visual union regions and relation predicates in the same semantic space.

**Strengths:**

1. It proposes UASAN to explicitly model the alignment between visual union regions and relation predicates, improving relation prediction.
2. Extensive experiments on VidVRD and VidOR datasets demonstrate significant performance improvements over existing methods.

**Limitations:**

1. As the open-vocabulary setting is not first proposed by this work, it is not suitable to illustrate the open-vocabulary setting in Figure 1.
2. While the paper demonstrates that a major contribution is that the UASAN consists of three cooperative components, I wonder what is the contribution of each component?
3. It would be better to provide some qualitative results to show that the model can really detect the visual union.
4 It should be highlighted why visual union can contribute to the open-vocabulary case.

**Suitability:**

3

---

### Meta-Review · Area_Chair_uM7d · 2024-06-26

**Recommendation:** Accept (Poster)
**Confidence:** 5

**Metareview:**

This paper initially received relatively positive reviews (2 Weak Accept, 1 Borderline Accept, 1 Borderline Reject). The main concerns are about the: 1) The presentation needs to be further improved (such as figures, experimental settings); 2) More comprehensive ablations on each component; 3) Insights behind the designs. After the rebuttal, most of these concerns have been addressed, and the Borderline Accept has been raised to Weak Accept. In total, we think this submission is meaningful for our community and we recommend Accept.